# Patient-Reported Sexual Function, Bladder Function and Quality of Life for Patients with Low Rectal Cancers with or without a Permanent Ostomy

**DOI:** 10.3390/cancers16010153

**Published:** 2023-12-28

**Authors:** Michael K. Rooney, Melisa Pasli, George J. Chang, Prajnan Das, Eugene J. Koay, Albert C. Koong, Ethan B. Ludmir, Bruce D. Minsky, Sonal S. Noticewala, Oliver Peacock, Grace L. Smith, Emma B. Holliday

**Affiliations:** 1Department of Gastrointestinal Radiation Oncology, The University of Texas MD Anderson Cancer Center, Houston, TX 77030, USA; mkrooney@mdanderson.org (M.K.R.); paslim20@students.ecu.edu (M.P.); ssnoticewala@mdanderson.org (S.S.N.);; 2Department of Colon and Rectal Surgery, The University of Texas MD Anderson Cancer Center, Houston, TX 77230, USA

**Keywords:** PROs, rectal cancer, ostomy, sexual function, bladder function, quality of life

## Abstract

**Simple Summary:**

This survey study investigates long-term patient-reported quality of life for individuals with low-lying rectal cancers, particularly focusing on the potential impact that ostomies may have on overall, sexual, and urinary quality of life. The findings suggest that patients with ostomies may experience a worse quality of life, affecting various aspects of daily life and relationships. These insights may help to inform patient counseling and shared decision making in the context of evolving rectal cancer treatment paradigms where patients have increasing multidisciplinary options.

**Abstract:**

Background: Despite the increasing utilization of sphincter and/or organ-preservation treatment strategies, many patients with low-lying rectal cancers require abdominoperineal resection (APR), leading to permanent ostomy. Here, we aimed to characterize overall, sexual-, and bladder-related patient-reported quality of life (QOL) for individuals with low rectal cancers. We additionally aimed to explore potential differences in patient-reported outcomes between patients with and without a permanent ostomy. Methods: We distributed a comprehensive survey consisting of various patient-reported outcome measures, including the FACT-G7 survey, ICIQ MLUTS/FLUTS, IIEF-5/FSFI, and a specific questionnaire for ostomy patients. Descriptive statistics and univariate comparisons were used to compared demographics, treatments, and QOL scores between patients with and without a permanent ostomy. Results: Of the 204 patients contacted, 124 (60.8%) returned completed surveys; 22 (18%) of these had a permanent ostomy at the time of survey completion. There were 25 patients with low rectal tumors (≤5 cm from the anal verge) who did not have an ostomy at the time of survey completion, of whom 13 (52%) were managed with a non-operative approach. FACTG7 scores were numerically lower (median 20.5 vs. 22, *p* = 0.12) for individuals with an ostomy. Sexual function measures IIEF and FSFI were also lower (worse) for individuals with ostomies, but the results were not significantly different. MLUTS and FLUTS scores were both higher in individuals with ostomies (median 11 vs. 5, *p* = 0.06 and median 17 vs. 5.5, *p* = 0.01, respectively), suggesting worse urinary function. Patient-reported ostomy-specific challenges included gastrointestinal concerns (e.g., gas, odor, diarrhea) that may affect social activities and personal relationships. Conclusions: Despite a limited sample size, this study provides patient-centered, patient-derived data regarding long-term QOL in validated measures following treatment of low rectal cancers. Ostomies may have multidimensional negative impacts on QOL, and these findings warrant continued investigation in a prospective setting. These results may be used to inform shared decision making for individuals with low rectal cancers in both the settings of organ preservation and permanent ostomy.

## 1. Introduction

Historically, the treatment of low rectal tumors within 5 cm of the anal verge includes abdominoperineal resection (APR) [1]. However, the widespread adoption of total mesorectal surgery techniques as well as ultralow low anterior resections have reduced the frequency with which APRs are performed [2]. Additionally, preoperative chemoradiation therapy (CRT) has proven useful to downstage low rectal tumors prior to surgery and allows for sphincter-sparing surgery (SSS) [3]. Despite these advances, it is estimated that approximately 40% of patients with rectal cancer undergo an APR and will have a permanent ostomy [4]. An ostomy refers to the exteriorization of the bowel through the abdominal wall and is a common procedure for individuals with gastrointestinal cancers requiring definitive or palliative surgery [5].

Disease-free survival is improving for patients with rectal cancer in the era of total neoadjuvant therapy (TNT) followed by surgical resection [6,7]. Aside from improved cancer cure rates, patients are increasingly interested in maintaining their quality of life (QOL) during survivorship [8]. When compared to SSS, patients undergoing APR are at increased risk for perineal wound complications, delayed healing, and increased hospital stay, which may not only delay adjuvant therapy, but may significantly impair short-term QOL [9]. Studies are mixed regarding the detrimental impact of global QOL with APR compared with SSS for low rectal cancer; some report worse QOL [10,11], others report better QOL [12,13], and most report no difference [14,15] for APR compared with SSS. Some studies suggest patients who undergo APR have more bothersome urinary symptoms than patients who undergo SSS [16], although others report no difference [14]. Data are more clear that patients who undergo APR have worse sexual function compared with those who undergo SSS; pain and body image issues related to the stoma are contributing factors as well as increased risk of autonomic pelvic nerve injury [14,17].

While not all patients may be a candidate for SSS due to tumor- or patient-specific factors, shared decision making is key for patients who do have surgical options [18]. Treatment decision making has become more complex in the era of non-operative management (NOM). The publication of the Organ Preservation in Patients with Rectal Adenocarcinoma (OPRA) trial suggested approximately 50% of well-selected patients with low rectal cancer may have a complete clinical response (cCR) to TNT and may be able to defer surgery [19]. Patient-reported functional outcomes were not published as part of the initial manuscripts, but single-institution studies suggest improved symptom-specific and QOL outcomes for patients treated with NOM compared with those who received surgery [20].

With increasing attention focused on the functional and QOL benefits of selective omission of CRT for low-risk patients [21], more data are needed regarding functional and QOL implications of utilizing CRT to either facilitate an SSS or omit surgery altogether. The aim of this brief report is to share patient-reported outcomes for patients with low rectal cancer treated at our institution with CRT followed by either APR, SSS, or NOM to evaluate for potential differences in urinary function, sexual function, and overall QOL between those with and without a permanent ostomy. Therefore, findings from this investigation could be used clinically to improve the quality of shared decision making for individuals considering various treatment strategies for low rectal tumors.

## 2. Materials and Methods

### 2.1. Survey Distribution and Data Collection

We received Institutional Review Board approval for this project (protocol 2020-0513). We contacted all consecutive patients who completed pelvic radiation for rectal adenocarcinoma at a large tertiary cancer center between 1 January 2017 and 31 December 2020 and were alive without evidence of disease recurrence. Patients eligible for this analysis had tumors ≤5 cm from the anal verge or had a permanent ostomy at the time of survey distribution. Patients all provided informed consent to participate in an online survey of validated patient-reported outcome measures (PROMs) including the Functional Assessment of Cancer Therapy-General (7-item version) (FACT-G7) survey [22], the International Consultation on Incontinence Questionnaire (ICIQ) Male Lower Urinary Tract Symptoms (MLUTS) [23] or Female Lower Urinary Tract Symptoms (FLUTS) questionnaire [24], the International Index of Erectile Function 5-item questionnaire (IIEF-5) [25], or the Female Sexual Function Index (FSFI) questionnaire [26]. Patients with a permanent ostomy also received the City of Hope Quality of Life-Ostomy Questionnaire [27]. The survey was administered using REDCap v13.11.2(©2013 Vanderbilt University) [28], and patients who returned completed surveys received a USD 10 Amazon gift card to show appreciation for their time.

Information about oncologic treatment was obtained from the medical record. All patients were discussed at a dedicated rectal cancer multidisciplinary conference with representation by colorectal surgeons, medical oncologists, radiologists, pathologists, and radiation oncologists. Radiation dose and fractionation were chosen at the discretion of the treating radiation oncologist with input from the multidisciplinary team. During this period at our institution, preoperative short-course radiation or long-course CRT was routinely recommended for patients with T3, T4, or node-positive low rectal cancer. If a complete clinical response was confirmed by endoscopy and MRI, non-operative management was discussed. If sufficient margins could be obtained with a low coloanal anastomosis after neoadjuvant treatment, SSS would be performed. Otherwise, patients would undergo APR.

### 2.2. Statistical Analysis

Descriptive statistics and frequency tables were used to summarize patient demographics. Pearson’s chi-square test and the Mann–Whitney U test were used to compare patient and treatment characteristics between patients treated for low rectal cancer with a permanent ostomy and those who did not receive a permanent ostomy. Descriptive statistics and frequency tables were also used to summarize FACT-G7, MLUTS or FLUTS, IIEF-5, or FSFI scores as well as answers to items on the City of Hope Quality of Life-Ostomy Questionnaire, when applicable. PROM scores were compared between patients with and without an ostomy at the time of survey completion using Pearson’s chi-square tests. Statistical analysis was performed using R 4.0.3 (R Foundation for Statistical Computing, Vienna, Austria).

## 3. Results

Of the 204 patients contacted, 124 (60.8%) returned completed surveys; 22 (18%) of these had an ostomy at the time of survey completion. There were 25 patients with low rectal tumors (≤5 cm from the anal verge) who did not have an ostomy at the time of survey completion, of whom 13 (52%) were managed with a non-operative approach. This cohort of 47 individuals was included for the primary analysis. Demographic, disease, and treatment characteristics of the study cohort are summarized in Appendix A, with results displayed separately for individuals with and without an ostomy at the time of survey completion. Overall, there were no significant differences between populations. Most respondents were non-Hispanic (80.9%) white (85.1%) men (66%). Most patients were treated with long course radiotherapy (78.7%) using a 3DCRT technique (61.7%). The median (first quartile Q1–third quartile Q3) time from completion of radiotherapy to survey completion was 34.2 months (20.3–49.5 months).

Distributions of PROs are summarized in Table 1, with results stratified by presence of an ostomy. Composite and subscore distributions of the FACT G7 score are displayed in Figure 1. Overall, individuals with an ostomy reported numerically worse FACT G7 composite scores, although differences were not statistically significant (*p* = 0.12). There were no differences in FACTG7 scores by sex (mean score for females and males were 20.7 and 20.6, respectively; *p* = 0.8). Sexual (IIEF, FSFI) and urinary-related (MLUTS, FLUTS) PROs are summarized in Appendix A, with results stratified by ostomy status. Composite IIEF and FSFI were numerically greater (indicating better sexual function) for men and women without an ostomy, although results were not significantly different (*p* = 0.18 and 0.83, respectively). MLUTS and FLUTS scores were higher for men and women with an ostomy (*p* = 0.06 and 0.01, respectively), indicative of worse urinary symptoms.

Survey responses related directly to ostomy and function and the impact on quality of life are summarized in Figure 2. Most patients reported significant impacts of the ostomy across myriad domains, including gastrointestinal concerns such as gas, odor, diarrhea, constipation, and pouch leakage. Many patients also responded that the ostomy had significant negative impacts on personal relationships and sex life. Further, most patients reported the ostomy negatively affecting their ability to participate in social and recreational activities.

## 4. Discussion

In this observational survey-based study, we investigated long-term patient-reported QOL for individuals with low rectal cancers treated with multimodality therapy, focusing particularly on the impact of permanent ostomies on function and QOL during long-term survivorship. We found that QOL was numerically worse for individuals with ostomies compared to those without, as measured by various instruments assessing general QOL, sexual function, and bladder function. Furthermore, patients with ostomies reported that the ostomies themselves often had a significant impact on their relationships and daily lives, including worry about odor and gas, and that they can be difficult to manage. The results of this study can be used to improve patient education and shared decision making for patients with low rectal cancers who may be candidates for various treatment options.

QOL following treatment of rectal cancer has been studied extensively. Most early reports focused on the impact of multimodality therapy on bowel, urinary, and sexual function, suggesting a multifaceted negative effect of treatment [29]. However, the treatment paradigm for rectal cancer has evolved significantly over the past decade with randomized evidence suggesting that select individuals may benefit from treatment de-escalation via omission of various modalities including radiotherapy and surgery [19,30]. These strategies may lead to improved patient QOL, and thus, efforts to compare long-term outcomes across approaches are critically needed. Initial reports from the PROSPECT trial (Alliance N1048) suggest important differences in QOL for those receiving neoadjuvant therapy consisting of CRT with fluorouracil compared with fluorouracil and oxaliplatin alone [21]. However, there are limited data to date comparing the impact that omission of surgery may have on QOL, and even fewer studies exist aiming specifically to understand the impact that ostomy placement may have on patient experience. As such, this investigation fills an unmet need by comparing outcomes for individuals with low rectal tumors that may be candidates for various de-escalation strategies including organ-preservation to avoid ostomy. Our center is a large tertiary referral center and thus is uniquely positioned to provide needed data in this understudied area.

The FACT G7 instrument has been utilized to study patient-reported QOL for individuals with cancer for over a decade and has been validated across numerous cancer types [31,32]. It includes seven broadly spanning questions related to daily functional and physical well-being with composite scores ranging from 0–28, with higher scores indicating better QOL. Prior research surveying over 400 patients with colorectal cancers showed a mean score of approximately 20, with more advanced disease status being associated with lower (worse) scores [33]. These results closely resemble the results from our study cohort (Table 1). Among respondents, those with ostomies tended to have lower scores overall, indicative of worse overall QOL, although results were not statistically significant (*p* = 0.12), possibly indicative of a limited sample size to detect differences across groups. Studies evaluating QOL after ostomy for patients with colorectal cancer found that living with a permanent ostomy impacts QOL negatively due to sexual problems, depressive feelings, gas, constipation, dissatisfaction with appearance, change in clothing, travel difficulties, feeling tired, and worry about noises [34,35,36,37]. Alternatively, our results may support published studies which suggest that the presence of a permanent ostomy does not have an adverse impact on overall QOL [38,39].

When assessing patient-reported sexual and urinary function via the IIEF and FSFI, and MLUTS and FLUTS, respectively, we found similar results, wherein individuals with ostomies tended to report worse function and QOL. IIEF scores range from 5 to 25 and FSFI scores range from 2 to 36, with higher scores reflecting better sexual function. In this study population, the median overall IIEF score was 14, with numerically higher scores in men without an ostomy; similarly, the median overall FSFI was 16.4 with lower scores in women with an ostomy, suggesting worse sexual function in both sexes. Unfortunately, not all individuals were sexually active when they completed the survey, and there were relatively few women in the study population. We were unable to show a statistically significant difference between groups as a result, likely due to this inadequate sample size, but these results are in line with other published results suggesting worse sexual function following permanent ostomy placement [40,41].

Patient-reported urinary dysfunction was less common in this cohort, with median MLUTS and FLUTS of 9.6 and 11.6, respectively. For reference, cutoff values of 12 or higher have been proposed to define moderate to severe urinary dysfunction [42]. Despite the small sample size, women with ostomies had significantly higher FLUTS scores (median 17 vs. 5.5, *p* = 0.01), indicating worse urinary dysfunction. Similarly, MLUTS scores were higher for men with ostomies (median 11 vs. 5.5, *p* = 0.06), but the results did not reach statistical significance despite a large numeric difference between the groups, likely owing to limited power. Taken together, these results corroborate prior research showing that urinary dysfunction is quite common for individuals with rectal cancers [43] and importantly suggest that permanent ostomy placement is a risk factor for worse urinary function.

Our results also provide valuable data regarding the real-world impact that an ostomy may have on daily lifestyle and activity based upon responses from the City of Hope Quality of Life-Ostomy Questionnaire. Many patients reported concern about gastrointestinal symptoms including worry about odor, gas, and diarrhea (Figure 2). Further, many individuals felt that their relationships with others and their ability to participate in recreational activities were affected by their ostomy. These data are critical to consider during the shared decision making and informed consent process for rectal cancer treatment [44]. NOM is becoming increasingly possible for select individuals [45], and our data lend support to studies such as the Dutch Watch-and-Wait Consortium which show some functional issues after definitive chemoradiation, but overall better QOL scores compared with those requiring total mesorectal excision [46].

Although this study draws strength from rigorous survey methodology to assess a holistic battery of patient-reported outcomes, it is limited by several factors related to experimental design. First, only a relatively small portion of the responding population had ostomies at the time of survey completion, and thus the study sample size was quite low, which limits our ability to detect differences between groups. Furthermore, because many patients received follow-up care at outside institutions, we were unable to perform a pre-specified power calculation to determine an ideal survey sample size for individuals with a permanent ostomy. Second, we did not have the ability to measure baseline data, so we may be incompletely capturing the true impact that an ostomy has on primary outcomes. Additionally, it is important to recognize that there are various approaches to ostomy placement that may impact patient-reported experience and QOL. For example, perineal colostomy refers to a procedure wherein the ostomy is connected to the perineum as opposed to the abdominal wall; prior research has suggested improved QOL with this approach [47]. In the present study, we did not attempt to investigate the impact of various ostomy procedures and thus our results may not be generalizable across all patients. Last, although we were unable to find any significant baseline differences between individuals with and without ostomies and had limited power to perform a matching approach across groups, it is possible that there were unaccounted factors that might affect the necessity for an ostomy and thus introduce potential confounding with survey responses. Nonetheless, these data raise important questions and contribute valuable information [47] in a relatively understudied area.

## 5. Conclusions

For individuals with low-lying rectal cancers, patient-reported QOL tended to be worse for individuals with ostomies compared to those without, as measured using various instruments assessing general QOL, sexual function, and bladder function. Ostomies themselves can be difficult to take care of and may affect a person’s relationships with others and his or her ability to participate in normal activities. The results of this study may be used to counsel patients who require an APR and may also be used to improve shared decision making for patients with low-lying rectal tumors who could be potential candidates at the time of treatment decision making for various treatment options, including organ preservation vs. permanent ostomy. In future studies, more data are needed, ideally collected in a multicenter prospective manner, to specifically compare relative functional and QOL issues between selective omission of radiation and definitive surgery versus the use of definitive chemoradiation and selective omission of surgery.

## Figures and Tables

**Figure 1 cancers-16-00153-f001:**
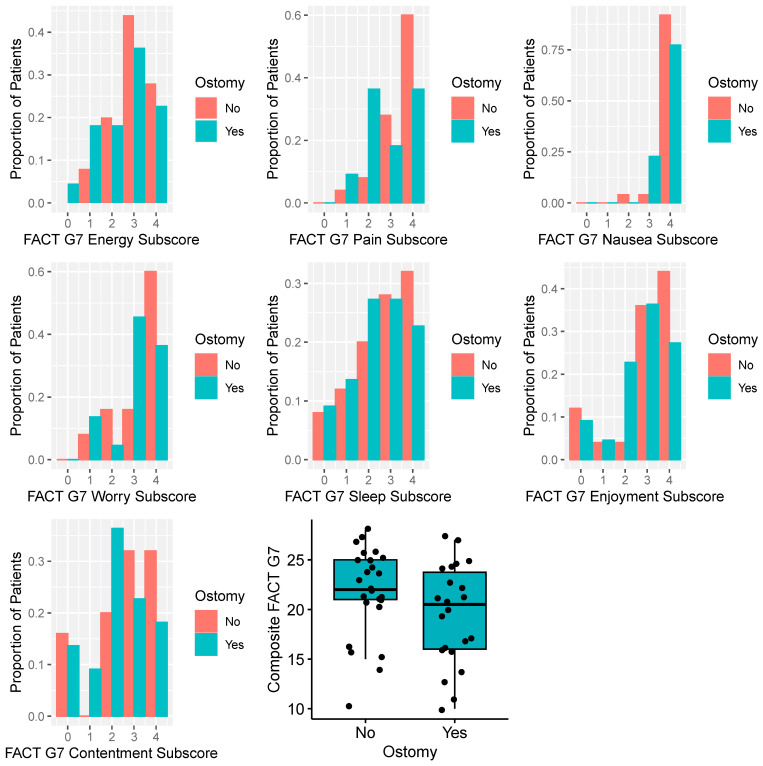
Distribution of Functional Assessment of Cancer Therapy General 7 (FACTG7) question instrument scores with results displayed separately by presence of an ostomy at survey completion. Higher scores reflect better quality of life.

**Figure 2 cancers-16-00153-f002:**
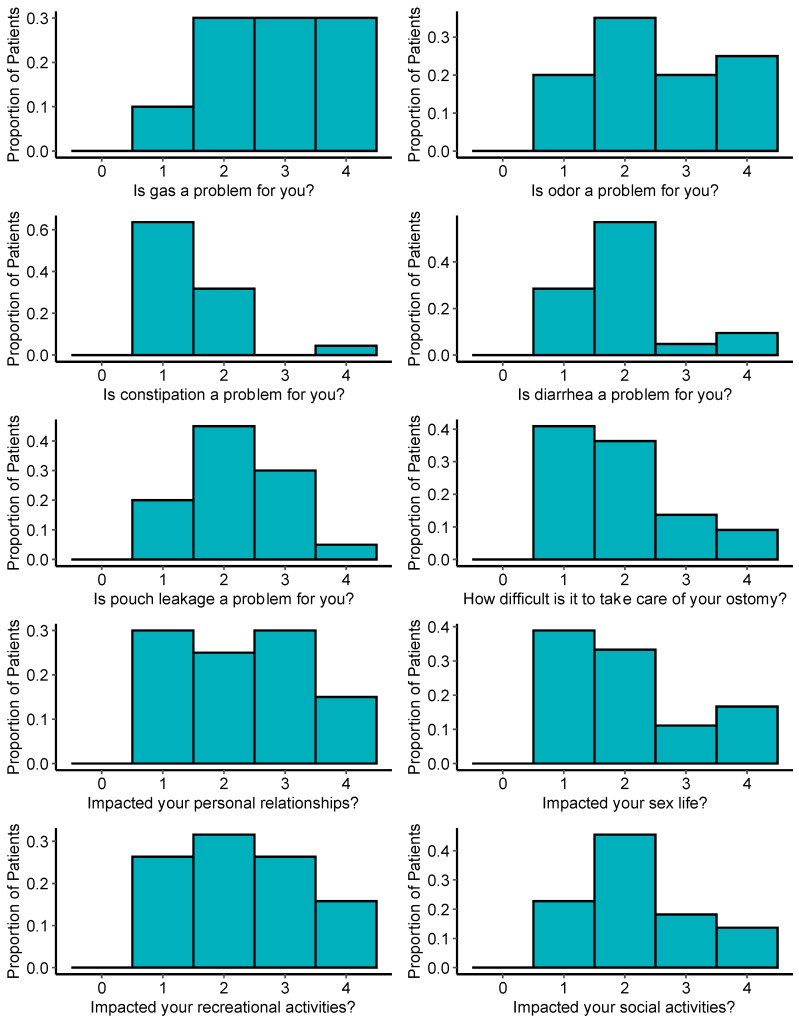
Distribution of survey responses related to ostomy function and impact on quality-of-life based on the City of Hope Quality of Life-Ostomy Questionnaire. Higher scores reflect greater symptom burden or impact on quality of life.

**Table 1 cancers-16-00153-t001:** Patient-reported outcomes with results displayed separately by presence of an ostomy at survey completion.

Score	Presence of a Permanent Ostomy			
	No	Yes	Overall	*p*-Value
FACTG7	*n* = 25	*n* = 22	*n* = 47	
Mean (SD)	21.8 (4.54)	19.5 (4.99)	20.7 (4.84)	0.12
Median [Q1, Q3]	22 [20.5, 25]	20.5 [16, 24]	21 [16, 25]	
IIEF	*n* = 17	*n* = 12	*n* = 29	
Mean (SD)	15.5 (6.37)	11.9 (7.33)	14 (6.89)	0.18
Median [Q1, Q3]	16 [9, 22]	8.5 [6.5, 19.5]	14 [7.5, 21]	
FSFI	*n* = 4	*n* = 6	*n* = 10	
Mean (SD)	18.3 (9.19)	15.1 (5.78)	16.4 (7.02)	0.83
Median [Q1, Q3]	18.6 [10.4, 26.15]	14.2 [10.5, 16.6]	14.2 [10.5, 25.4]	
MLUTS	*n* = 18	*n* = 12	*n* = 30	
Mean (SD)	6.50 (4.66)	14.3 (13.7)	9.60 (9.92)	0.06
Median [Q1, Q3]	5.50 [3, 8]	11.0 [4.5, 17]	6.50 [3, 13]	
FLUTS	*n* = 6	*n* = 9	*n* = 15	
Mean (SD)	5.33 (3.27)	15.8 (7.92)	11.6 (8.23)	0.01
Median [Q1, Q3]	5.50 [3, 7]	17 [8.5, 22]	10 [4, 18]	

Abbreviations: FACTG7 = Functional Assessment of Cancer Therapy—General 7 Question Survey; IIEF = International Index of Erectile Function; FSFI = Female Sexual Function Index; MLUTS = Male Lower Urinary Tract Symptom score; FLUTS = Female Lower Urinary Tract Symptom Score.

## Data Availability

Data will be made available upon reasonable request to the study authors.

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
