# Peer review of "Patient-Reported Sexual Function, Bladder Function and Quality of Life for Patients with Low Rectal Cancers with or without a Permanent Ostomy"

_cancers, 2023, doi:10.3390/cancers16010153_

Round 1
Reviewer 1 Report
Comments and Suggestions for Authors
Authors are to be commended to take on this important research question in a relatively understudied area. Due to limited sample size and despite rigorous survey methodology clear answers cannot be given. In a prospective multicentre study setting sufficient patient numbers could be accrued.
Reviewer 2 Report
Comments and Suggestions for Authors
1- More literature is required and also what does it add to the subject area compared with other published material?
2- What makes their work novel in this instance? In the paper, the novelty needs to be defined precisely.
3- As a research article, I think the authors need to add more figures in the results section.
4- All abbreviations used in figures and tables should be defined in their legends.
5- In the Discussion section, a comparison of the work with the previous literature results are needed.
6- There are a few grammatical errors that authors should double-check in the whole manuscript.
Comments on the Quality of English LanguageThere are a few grammatical errors that authors should double-check in the whole manuscript.
Reviewer 3 Report
Comments and Suggestions for Authors
An interesting study that evaluates PROs of patients with low rectal cancers. Some comments:
-Was a sample size estimation performed? Overall, the number of analyzed patients is small and thus the authors should consider extending the screening period.
-Demographics of included patients should be included in the main text and not supplementary material.
-Please support the novelty of this study and add an extensive strengths paragraph.
-Was a matching algorithm performed between the study groups?
Reviewer 4 Report
Comments and Suggestions for Authors
This manuscript introduces and comprehensively discusses the overall, sexual-, and bladder-related patient-reported quality-of-life (QOL) for individuals with low rectal cancers as well as explores potential differences in patient-reported outcomes between patients with and without a permanent ostomy
-The topic is original and relevant to the field. There is limited information on this topic in the literature.
-This article makes clear that ostomies may have multidimensional negative impacts on QOL, and these findings warrant continued investigation in a prospective setting.
There are no further improvements regarding the methodology.
-The conclusions are consistent with the evidence and arguments presented as well as summarize the main point of this article.
-References are up-to-date and appropriate
-Tables and figures are well formatted and make the study easy to follow
Minor revision
1)"The word stoma or ostomy is derived from the Latin word ostium, which means opening or mouth. An intestinal stoma is one of the most common surgical procedures. The exteriorization of either the small bowel (ileostomy) or large bowel (colostomy) through the anterior abdominal wall is performed. The first recorded intestinal stoma was created by the German surgeon Baum in 1879 to divert an obstructing colon carcinoma."
Add this information to the introduction section and consider citing:
https://www.ncbi.nlm.nih.gov/books/NBK565910/
2) I would like a brief discussion on LARS in patients who underwent sphincter-saving resection
3) "The perineal colostomy is a reconstruction method performed after abdominoperineal resection for rectal malignancy. In this technique, the permanent colostomy is not placed in the left quadrant of the abdomen, but in the perineum. According to the literature, this technique provides many advantages such as a higher degree of satisfaction and greater quality of life to patients."
I would like a brief discussion on the advantages and disadvantages of this technique.
Is this an alternative method to promote patients' satisfaction and safety?
Reviewer 5 Report
Comments and Suggestions for Authors
In this study, the authors investigated long-term patient-reported quality of life for individuals with low-lying rectal cancers, by a comprehensive survey consisting of various patient-reported outcome measures. They found that FACTG7 scores were numerically lower for individuals with an ostomy. Sexual function measures IIEF and FSFI were also lower (worse) for individuals with ostomies, along with worse urinary function. The conclusions were supported by some solid data. However, a few issues need to be addressed.
1. Did the authors try to know whether there was differences between female and male patients for the quality-of-life with ostomy?
2. For the introduction part, it is not strong enough to attract readers.
3. It is better to have abbreviation list.
Comments on the Quality of English LanguageEnglish language is fine.
Round 2
Reviewer 2 Report
Comments and Suggestions for Authors
The authors corrected all of referee points and in this form it is acceptable for further procedure of publication.